

# The reliability of vertical jump tests between the Vertec and *My Jump* phone application

Vanessa R. Yingling, Dimitri A. Castro, Justin T. Duong, Fiorella J. Malpartida, Justin R. Usher and Jenny O

Department of Kinesiology, California State University, East Bay, Hayward, CA, USA

Corresponding author
Vanessa R. Yingling,
vanessa.yingling@csueastbay.edu

## ABSTRACT

**Background:** The vertical jump is used to estimate sports performance capabilities and physical fitness in children, elderly, non-athletic and injured individuals. Different jump techniques and measurement tools are available to assess vertical jump height and peak power; however, their use is limited by access to laboratory settings, excessive cost and/or time constraints thus making these tools oftentimes unsuitable for field assessment. A popular field test uses the Vertec and the Sargent vertical jump with countermovement; however, new low cost, easy to use tools are becoming available, including the *My Jump* iOS mobile application (app). The purpose of this study was to assess the reliability of the *My Jump* relative to values obtained by the Vertec for the Sargent stand and reach vertical jump (VJ) test.

**Methods:** One hundred and thirty-five healthy participants aged 18–39 years (94 males, 41 females) completed three maximal Sargent VJ with countermovement that were simultaneously measured using the Vertec and the *My Jump*. Jump heights were quantified for each jump and peak power was calculated using the Sayers equation. Four separate ICC estimates and their 95% confidence intervals were used to assess reliability. Two analyses (with jump height and calculated peak power as the dependent variables, respectively) were based on a single rater, consistency, two-way mixed-effects model, while two others (with jump height and calculated peak power as the dependent variables, respectively) were based on a single rater, absolute agreement, two-way mixed-effects model.

**Results:** Moderate to excellent reliability relative to the degree of consistency between the Vertec and *My Jump* values was found for jump height (ICC = 0.813; 95% CI [0.747–0.863]) and calculated peak power (ICC = 0.926; 95% CI [0.897–0.947]). However, poor to good reliability relative to absolute agreement for VJ height (ICC = 0.665; 95% CI [0.050–0.859]) and poor to excellent reliability relative to absolute agreement for peak power (ICC = 0.851; 95% CI [0.272–0.946]) between the Vertec and *My Jump* values were found; Vertec VJ height, and thus, Vertec calculated peak power values, were significantly higher than those calculated from *My Jump* values ($p < 0.0001$).

**Discussion:** The *My Jump* app may provide a reliable measure of vertical jump height and calculated peak power in multiple field and laboratory settings without the need of costly equipment such as force plates or Vertec. The reliability relative to degree of consistency between the Vertec and *My Jump* app was moderate to excellent. However, the reliability relative to absolute agreement between Vertec and *My Jump* values contained significant variation (based on CI values), thus, it is

recommended that either the *My Jump* or the Vertec be used to assess VJ height in repeated measures within subjects' designs; these measurement tools should not be considered interchangeable within subjects or in group measurement designs.

## INTRODUCTION

Vertical jump height is a measurement that coaches, physical educators, health care professionals, and strength and conditioning practitioners use to predict or assess physical performance for talent identification and player development purposes. For example, many sport scouting combines use vertical jump performance to identify talent (football and basketball) (*Teramoto, Cross & Willick, 2016*). Moreover, the literature has demonstrated that individual and sport characteristics such as sex, skill level, sport position, and risk of injury are associated with vertical jump performance (*Burr et al., 2008*; *Mujika et al., 2009*; *Marques & Izquierdo, 2014*; *Janot, Beltz & Dalleck, 2015*; *Spiteri et al., 2017*). Vertical jump performance has also been found to be associated with neuromuscular fatigue and thus has been used to monitor and avoid overtraining in athletes (*Gathercole et al., 2015*). Lastly, vertical jump tests correlate with total and lower extremity lean mass (*Stephenson et al., 2015*) and bone strength (*Janz et al., 2015*; *Yingling et al., 2017*). The "gold standards" for vertical jump height measurement are video analysis to calculate the position of the body's center of mass (*Aragón, 2000*) and integration of the ground reaction force measured on a force plate (*Menzel et al., 2010*). However, relative to "real-world" assessment by non-elite and/or non-research populations, limited access to laboratory settings, excessive cost of such measurement tools, time, and/or expertise constraints render these approaches largely unsuitable for field assessments conducted by many sport and physical activity practitioners.

Many devices have been developed to measure vertical jump height in a low cost and reliable manner, including contact mats (Just Jump System, Ergo Jump), velocity systems (GymAware, accelerometers), and linear position transducers (OptoJump, Myotest, Vertec). Three factors can affect the reliability and validity of all these approaches: the method used to calculate height, the type of jump performed, and body mass. The force plate, considered the "gold standard," measures jump height by calculating flight time of the jump (*Walsh et al., 2006*; *Glatthorn et al., 2011*); however, excessive hip and/or knee flexion during the jump can overestimate flight time and jump height (*Nuzzo, Anning & Scharfenberg, 2011*). The type of jump used to assess subjects has varied between studies and was typically dependent on the purpose of the assessment, the population assessed, and the setting of the assessment. The squat jump (SJ) and countermovement jump (CMJ) are predominantly used in laboratory settings (*Markovic et al., 2004*; *Nuzzo, Anning & Scharfenberg, 2011*), but a common field test used in physical education settings as well as in professional sport combines is the Sargent jump and reach test (VJ) (*De Salles et al., 2012*; *Castagna et al., 2013*; *Ayán-Pérez et al., 2017*). The VJ is not only

focused on the lower limbs but necessitates coordination of both the lower and upper limbs (*Markovic et al., 2004*; *Leard et al., 2007*; *Nuzzo, Anning & Scharfenberg, 2011*) as the upper limbs may increase take-off velocity up to 10% (*Luhtanen & Komi, 1978*; *Harman et al., 1988*). High reliability of the jump and reach test was reported in both pre-school age children (*Ayán-Pérez et al., 2017*) and in athletes (14-year-old soccer players) (*De Salles et al., 2012*). In addition, ecological validity was found for VJ testing in activities such as basketball and volleyball, in which reaching height is key during the jump (*Menzel et al., 2010*). A common measuring tool of the VJ in the field is the Vertec, however the Vertec requires execution of a more complex jump from the participant, potentially affecting the reliability of the VJ heights (*Buckthorpe, Morris & Folland, 2012*; *Harman et al., 1988*; *Nuzzo, Anning & Scharfenberg, 2011*). The participant must be able to coordinate the arm swing such that the arms are fully extended and in contact with the vanes at the moment that the participant has attained their greatest displacement from the floor (*Harman et al., 1988*). In addition, measurement error may be introduced due to the two-step measurement protocol for the Vertec as well as the vane spacing on the Vertec (*Nuzzo, Anning & Scharfenberg, 2011*). Yet, the use of the Vertec to measure VJ height remains commonplace in physical education and sport settings due to its convenience and price point. Although the measurement tool and type of jump may introduce error, body mass is also a large factor affecting VJ height (*Sayers et al., 1999*). The difference in body mass can significantly affect the vertical height reached by two participants yet the impulse generated may be similar (*Harman et al., 1988*).

A new approach to vertical jump height measurement is the use of mobile applications. *My Jump*, a mobile application for iOS and android devices, uses the device camera's frame-by-frame analysis to calculate flight time and jump height. Recent studies have found almost perfect agreement between the force plate and *My Jump* for measuring CMJ height using either time in air (*Balsalobre-Fernández, Glaister & Lockey, 2015*; *Driller et al., 2017*) or calculated height from take-off velocity (*Carlos-Vivas et al., 2018*). Furthermore, excellent agreement between force plate and *My Jump* measurements was found for three different types of jumps including the CMJ, SJ and drop jump (DJ) in both male and female competitive athletes (*Gallardo-Fuentes et al., 2016*; *Stanton, Wintour & Kean, 2017*). Intra rater reliability for both CMJ and DJ was also found to be excellent (*Stanton, Wintour & Kean, 2017*). *My Jump* is an affordable, portable alternative to other tools that assess vertical jump performance. Moreover, high reliability and accuracy of *My Jump* compared to the gold standard (force plate) has been reported (*Balsalobre-Fernández, Glaister & Lockey, 2015*; *Gallardo-Fuentes et al., 2016*; *Carlos-Vivas et al., 2018*; *Stanton, Wintour & Kean, 2017*). However, both the force plate and *My Jump* use flight time as the source of the height calculation (*Balsalobre-Fernández, Glaister & Lockey, 2015*) while the commonly used field measurement is a direct distance measurement of jump height. Thus, the Vertec may yield different absolute jump height values compared to *My Jump,* yet no studies to date have compared *My Jump* to the Vertec. Therefore, the primary purpose of the study was to examine the reliability of *My Jump* VJ values compared to those of Vertec. A secondary purpose was to examine whether the use of raw VJ values versus calculated external peak power values using the
**Table 1 Participant characteristics.**

|  | Male ($n$ = 94) | Female ($n$ = 41) |
|---|---|---|
| Age (years) | 18–29 | 18–39 |
| Height (m) | 1.77 (0.08) | 1.67 (0.08) |
| Mass (kg) | 72.8 (9.9) | 63.5 (9.3) |

**Note:**

Age range and average height and mass of the male ($n$ = 94) and female ($n$ = 41) participants. Mean (SD).

Sayers equation influenced reliability results. We hypothesized that: (a) reliability relative to degree of consistency between the measurement tools (Vertec and *My Jump*) would be high, and (b) reliability, in terms of absolute agreement between the measurement tools would be low, with a detectable systematic difference.

# MATERIALS AND METHODS

## Correlational study

### *Participants*

One hundred and thirty-five healthy adults (94 males, 41 females; university students, staff, and faculty) participated in the study (Table 1). All participants were informed of the risks and benefits of the study and provided written informed consent. All study procedures were approved by the California State University, East Bay Institutional Review Board (CSUEB-IRB-2015-275-F).

### *Experimental Protocol*

*Procedures*

Participants completed a general health and demographic survey and were excluded if they had a history of health concerns, a disease or physical condition that may affect physical activity, or, were pregnant. The demographic information collected includes sex, height, and mass. Height and mass were measured using a stadiometer and a calibrated scale. All participants were asked if they were competitive athletes (yes/no; defined as: "One who plays an organized sport for a team or in an organization"), and whether they regularly participated in vigorous physical activity (yes/no; defined as: "Activity that causes large increases in breathing or heart rate for at least 10 min continuously").

Two vertical jump measuring systems, Vertec and *My Jump*, were used simultaneously to assess VJ height. "Peak Power" was then calculated from the jump height measured from the two measuring systems (*Sayers et al., 1999*). Jump height was quantified using a Vertec (JUMPUSA.com, Sunnyvale, CA, USA) while also being recorded using an iPad mini 2 (Frame Rate 60 fps, 1080 p video, Apple Inc, Cupertino, CA, USA). The take-off and landing frames from the video were determined using *My Jump* and flight time (ms) was then calculated. The jump height was determined using the calculation (*Bosco, Luhtanen & Komi, 1983*):

$$\text{Height(m)} = (g \times \text{time}^2)/8 \quad \text{where } g = 9.81 \text{ m/s}^2$$

The iPAD mini 2 was connected to a tripod and placed perpendicular to the frontal plane of the participants focused on their feet and approximately 1.5 m from the participant. One researcher was responsible for all analysis of flight time duration; takeoff was determined as the first frame with both feet off the ground and landing when at least one foot touched the ground.

Participants were given the option to participate in warm up exercises consisting of 10 squats, 10 alternating high knees, and 1 min running in place. Following verbal explanation of the jump and reach CMJ and a physical demonstration by a research assistant, the participants standing reach height was measured using the Vertec followed by three VJ jumps as high as possible to displace the Vertec vanes. At the moment preceding the jump, the participants could freely flex the hip, knee, and ankle joints and prepare the upper limbs for a sudden upward thrust, in an effort to promote the highest vertical jump possible. The rest time between jumps was 20 s. The participant's vertical jump height was calculated as the difference between their maximum jump height and standing reach height. "Peak Power" was calculated from the maximal jump height of three trials.

Sayers Peak Power Equation (*Sayers et al., 1999*)

$$\text{Peak power}(W) = [51.9 \times \text{VJ height(cm)}] + [48.9 \times \text{Body mass(kg)}] - 2007$$

## Statistical analysis

Intraclass correlation (ICC) is a measure of reliability which assesses both, degree of correlation (i.e., consistency) and degree of absolute agreement between two variables (*Shrout & Fleiss, 1979*). Given the purpose of our study, we were equally interested in consistency and absolute agreement between the two measurement tools. We were also interested in determining whether VJ jump height and peak power (calculated using the *Sayers et al. (1999)* equation) produced differential ICC results. As such, reliability was assessed using four separate ICC estimates and their 95% confidence intervals (calculated using SPSS statistical package version 23; SPSS Inc., Chicago, IL, USA). More specifically, we conducted four separate single rater two-way random-effects model ICCs. Two analyses (with jump height and peak power as the dependent variables, respectively) were based on a single rater, consistency, two-way random-effects model, while two others (with jump height and peak power as the dependent variables, respectively) were based on a single rater, absolute-agreement, two-way random-effects model. A two-way random-effects model is noted to be appropriate for evaluating assessment methods that are intended for routine use by raters with similar characteristics (*Koo & Li, 2016*). We chose a single rater ICC type as we assumed a single rater would be the basis for real world measurement of jump height (e.g., a single coach, trainer, PE teacher, etc. will administer the vertical jump test during assessment). We chose to adopt Koo and Li's guidelines for interpretation of ICC values; based on a confidence interval (CI) of 95% of the ICC estimate, <0.50, 0.50–0.75, 0.75–0.90, and >0.90 represent poor, moderate, good, and excellent ICC, respectively (*Koo & Li, 2016*). Analysis separating the participants by sex was also run. Paired student's *t*-test were performed to determine any systematic

**Table 2 Reliability between *My Jump* and Vertec.**

| | *My Jump* | Vertec | ICC(3,1) (95%CI) consistency | | ICC(3,1) (95%CI) absolute agreement | |
|---|---|---|---|---|---|---|
| **All Participants** | | | | | | |
| Vertical jump height (cm) | 43.05 (12.13) | 51.93 (14.36)* | 0.813 | 0.747–0.863 | 0.665 | 0.050–0.859 |
| Peak power (W) | 3,974 (1,043) | 4,435 (1,144) | 0.926 | 0.897–0.947 | 0.851 | 0.272–0.946 |
| **Females (*n* = 41)** | | | | | | |
| Vertical jump height (cm) | 31.52 (8.22) | 37.02 (9.36)* | 0.555 | 0.302–0.735 | 0.469 | 0.118–0.699 |
| Peak power (W) | 2,952 (684) | 3,238 (706)* | 0.807 | 0.667–0.892 | 0.748 | 0.425–0.881 |
| **Males (*n* = 94)** | | | | | | |
| Vertical jump height (cm) | 48.07 (9.94) | 58.43 (10.90)* | 0.732 | 0.623–0.814 | 0.492 | 0.086–0.774 |
| Peak power (W) | 4,420 (840)* | 4,958 (874)* | 0.893 | 0.844–0.928 | 0.747 | 0.007–0.913 |

**Notes:**
Interclass correlation values comparing the consistency and absolute agreement of the *My Jump* and Vertec for vertical jump height (cm) and peak power (W). Mean (SD).
* $p < 0.05$ paired *t*-test.

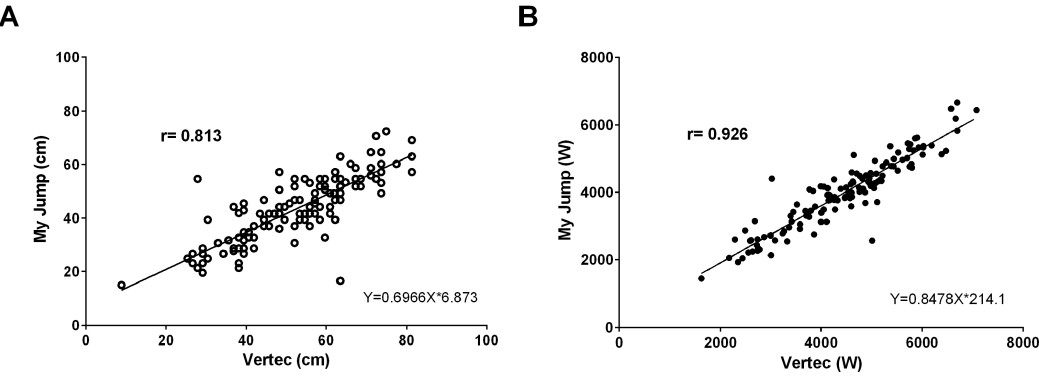

**Figure 1 Correlation between *My Jump* and Vertec.** (A) Vertical jump height (cm) $r = 0.813$. (B) Peak power (W) $r = 0.926$.               

differences between the absolute values of jump height between the two measurement tools, Vertec and *My Jump*. Bland–Altman plots were constructed using Graph Pad (GraphPad Prism version 6.00 for Windows, GraphPad Software, San Diego, CA, USA).

## RESULTS

### Consistency

The ICC estimate and 95% CI demonstrated good reliability for jump height (ICC = 0.813; 95% CI [0.747–0.863]) and excellent reliability for calculated peak power (ICC = 0.926; 95% CI [0.897–0.947]) between the Vertec and *My Jump*. These ICC results indicate that the Vertec and *My Jump* are highly consistent with each other with respect to measurement of maximum VJ height (Table 2; Figs. 1A and 1B). Furthermore, given the greater ICC estimate and greater and narrower CI for peak power values, our results indicate that the use of calculated peak power as the dependent variable resulted in stronger reliability values compared to VJ height. The ICC estimate and 95% CI demonstrated poor to moderate reliability for jump height depending on sex; males (*n* = 94) (ICC = 0.732; 95% CI [0.623–0.814]) and females (*n* = 41) (ICC = 0.555;

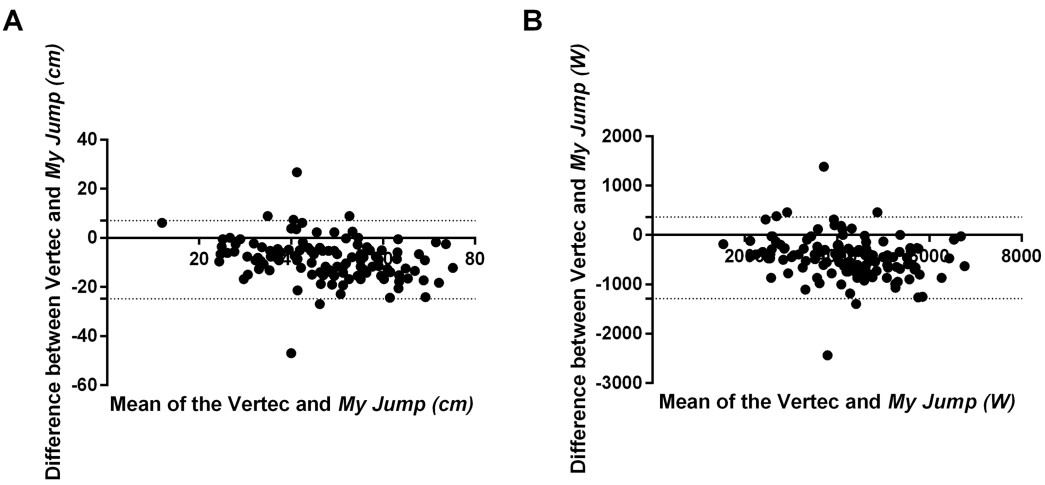

**Figure 2 Bland–Altman plots depicting the level of agreement in both.** (A) Maximal vertical jump height (cm) and (B) calculated peak power (W). The majority of points below the line of identity (average values of Vertec and *My Jump*) confirm the lower values using the *My Jump* compared to the Vertec.

95% CI [0.302–0.735]). Consistency for calculated peak power values increased for both groups to moderate to excellent; males (ICC = 0.893; 95% CI [0.844–0.928]) and females (ICC = 0.807; 95% CI [0.667–0.892]) (Table 2).

### Agreement

ICC estimates and 95% CI demonstrated poor to good reliability for jump height (ICC = 0.665; 95% CI [0.050–0.859]) and poor to excellent reliability for calculated peak power (ICC = 0.851; 95% CI [0.272–0.946]). Despite reasonable ICC estimates—particularly, for calculated peak power—the very broad CI for each dependent variable indicate that the Vertec and *My Jump* do not consistently produce similar absolute VJ height values relative to each other. A paired-samples *t*-test confirmed the lack of absolute agreement between the tools; mean VJ height using the Vertec (51.93 ± 14.36 cm) were found to be significantly higher (statistically) than mean VJ height values measured using *My Jump* (43.05 ± 12.13 cm; $t(134) = 12.69$, $p < 0.0001$; Table 2; Fig. 2). ICC estimates and 95% CI demonstrated poor to good reliability for jump height depending on sex; males (ICC = 0.492; 95% CI [0.086–0.774]) and females (ICC = 0.469; 95% CI [0.118–0.699]) and poor to excellent reliability for calculated peak power; males (ICC = 0.747; 95% CI [0.007–0.913]) and females (ICC = 0.748; 95% CI [0.425–0.881]). The very broad CI for each dependent variable indicate that the Vertec and *My Jump* do not consistently produce similar absolute VJ height values relative to each other for both males and females (Table 2).

### DISCUSSION

*My Jump* compared to Vertec demonstrated good to excellent reliability relative to degree of consistency, and poor to excellent reliability relative to absolute agreement. The force plate may be considered the "gold standard" for vertical jump testing accuracy (*Menzel et al., 2010*), however, this measurement tool is not easily accessible to non-elite and/or

non-professional physical activity practitioners due to environmental, financial, time, and/or expertise constraints (and thus, not commonly used by this population). Although preliminary support for the use of *My Jump* by field practitioners has been established (*Balsalobre-Fernández, Glaister & Lockey, 2015*; *Gallardo-Fuentes et al., 2016*; *Driller et al., 2017*), these reliability studies have compared *My Jump* to force plate data. Given that relatively few field practitioners are using force plates to measure vertical jump height and that the goal of applied research is to provide data and recommendations that are likely to be adopted by practitioners, it was important to examine the reliability of *My Jump* compared to a more commonly used field measurement tool. Like the force plate, the Vertec has also been found to be a reliable vertical jump measurement tool (*Klavora, 2000*; *Caruso et al., 2010*; *Nuzzo, Anning & Scharfenberg, 2011*), but unlike the force plate, the Vertec is amenable to multiple testing locations (e.g., laboratory, field, court, etc.) and thus, is more commonly used in "real-world" vertical jump test settings.

In a similar vein, it was important to examine the reliability of *My Jump* using a jump that most closely approximates the manner in which individuals actually perform maximum vertical jumps in the real world. Indeed, previous *My Jump* reliability studies have employed less ecologically valid jump styles (i.e., CMJ, SJ, and DJ) (*Balsalobre-Fernández, Glaister & Lockey, 2015*; *Gallardo-Fuentes et al., 2016*; *Driller et al., 2017*), thus reducing the generalizability of their findings to the real world. The VJ jump is not without criticism from an experimental control perspective; these criticisms have centered upon two issues: the complexity of the movement, and human measurement error (*Luhtanen & Komi, 1978*; *Harman et al., 1988*; *Leard et al., 2007*; *Menzel et al., 2010*; *Nuzzo, Anning & Scharfenberg, 2011*).

Reliability relative to absolute agreement between the jump height measurement tools ranged from poor to excellent (Figs. 1 and 2; Table 2) and the absolute jump height values measured via Vertec were significantly higher than those measured via *My Jump* (Fig. 2; Table 2). Thus, the data from this study indicates that the Vertec and *My Jump* do not consistently produce similar absolute jump height values relative to each other. These differences are due to the way in which jump height was calculated; the *My Jump* was based on time in the air and does not account for the upper limb reach component of the jump that was measured by the Vertec (*Menzel et al., 2010*; *Nuzzo, Anning & Scharfenberg, 2011*). This finding (a lack of absolute agreement between measurement tools) parallels that found in previous studies examining vertical jump heights in healthy adult participants (*Hoffman & Kang, 2002*; *Caruso et al., 2010*; *Menzel et al., 2010*). Collectively, based on these findings the recommendation is that field practitioners explicitly use either the *My Jump* or the Vertec to assess VJ jump height; one tool should be used exclusively for repeat measures and the measurement tools should not be considered interchangeable.

Reliability relative to degree of both consistency and absolute agreement increased for the calculated peak power values compared to jump height measures. The absolute differences in measurements between Vertec and *My Jump* were smaller when peak power was calculated from jump height (Table 2). Peak power calculations (*Sayers et al., 1999*) include body mass and body mass significantly affects an individual's ability to jump. Individuals with similar jump heights can have very different peak power values due to
body mass differences (*Harman et al., 1988*; *Johnson & Bahamonde, 1996*). One limitation of using peak power in the current study, which should be noted, is that the use of the Sayers regression equation to estimate peak power inherently inflated the ICC estimates due to the shared variance of body weight. Thus, the reader may wish to interpret the specific ICC values with caution. However, the reader can still confidently conclude that the use of peak power values demonstrated higher reliability values compared to that observed of jump height values.

From an ecological validity perspective, the specific jump style employed, the use of healthy adult participants from across the general university population, the relatively large number of participants, and our decision to test in the field rather than in a controlled laboratory space all represent strengths of the current study. Such data collection methods increase the generalizability of the current results. A possible limitation of our study was that some participants may not have been familiar with the VJ jump style. If that were the case for some participants, their resultant jumps may have been inconsistent from jump to jump, or, may not have be representative of their "true" maximum vertical jump height. In this study, we aimed to minimize the influence of this limitation and between jump reliability by providing verbal instructions and physically demonstrating the VJ jump style to participants prior to their VJ jump attempts, as well as by taking each participant's highest VJ of their three jump trials. Furthermore, success in the VJ tests requires a maximal impulse to be applied to the ground resulting in maximal momentum of the body (maximal vertical velocity). The net vertical impulse is highly correlated with external power flow to the ground (*Knudson, 2009*; *Winter et al., 2016*) resulting in the misuse of "power" associated with vertical jump success. The vertical jump test provides information about the ability of an individual to jump vertically, and the height of the jump should not be interpreted (through a predictive equation based on the height or the flight time of a jump or a series of jumps) as a true measure of the power an individual can generate but more as a measure of the jumping ability of an individual (*Tessier et al., 2013*). External power flow is a poor descriptor of performance compared with the impulse that changes velocity (*Knudson, 2009*; *Winter et al., 2016*), however we used the Sayer's equation to compare the two methods and used the terminology from that paper "Peak Power."

## CONCLUSION

Although Vertec and *My Jump* were found to be comparable tools for measuring VJ jump height, the relative ease of use, affordability, and portability makes *My Jump* an attractive option for non-elite and/or non-professional movement practitioners. However, practitioners should be aware that absolute VJ jump values for Vertec and *My Jump*, respectively, will differ significantly from each other (although the use of the "Peak Power" regression equation that includes body weight will minimize this difference). Thus, regardless of whether the practitioner chooses to use Vertec or *My Jump*, it is recommended that the selected tool should be used exclusively during repeated measures within-subject testing of individuals or groups.

## ACKNOWLEDGEMENTS

We thank Cal State East Bay's Center for Student Research for providing support and research development opportunities for our student researchers. We also thank Cal State East Bay's Kinesiology Research Group student researchers who assisted with data collection for the project.

### Funding

This work was supported by the California State University East Bay Center of Student Research (Faculty Support Grant for Mentoring Student Researchers, #FSG002). The funders had no role in study design, data collection and analysis, decision to publish, or preparation of the manuscript.

### Grant Disclosures

The following grant information was disclosed by the authors:
Faculty Support Grant for Mentoring Student Researchers: #FSG002.

### Competing Interests

The authors declare that they have no competing interests.

### Author Contributions

- Vanessa R. Yingling conceived and designed the experiments, analyzed the data, prepared figures and/or tables, authored or reviewed drafts of the paper, approved the final draft.
- Dimitri A. Castro performed the experiments, analyzed the data, prepared figures and/or tables, authored or reviewed drafts of the paper, approved the final draft.
- Justin T. Duong performed the experiments, analyzed the data, prepared figures and/or tables, authored or reviewed drafts of the paper, approved the final draft.
- Fiorella J. Malpartida performed the experiments, analyzed the data, authored or reviewed drafts of the paper, approved the final draft.
- Justin R. Usher performed the experiments, analyzed the data, authored or reviewed drafts of the paper, approved the final draft.
- Jenny O conceived and designed the experiments, analyzed the data, prepared figures and/or tables, authored or reviewed drafts of the paper, approved the final draft.

### Human Ethics

The following information was supplied relating to ethical approvals (i.e., approving body and any reference numbers):

All study procedures were approved by the California State University, East Bay Institutional Review Board (CSUEB-IRB-2015-275-F).

### Data Availability

The data set used for ICC analysis is provided as a Supplemental Dataset File.

## Supplemental Information

Supplemental information for this article can be found online at http://dx.doi.org/10.7717/peerj.4669#supplemental-information.

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
