# Peer review of "The reliability of vertical jump tests between the Vertec and My Jump phone application"

_PeerJ, doi:10.7717/peerj.4669_

## Round 0.1 · original submission · Major Revisions

Dear authors

Please see the comments of the reviewers. We look forward to receiving your revised manuscript.

·

Basic reporting

The authors present a well-written study on the reliability of MyJump vs. Vertec during a vertical jump. I have a number of concerns that, upon being addressed, can improve improve the quality of the paper.

General Comments
- Because consistency is high but absolute agreement is low, I think the reader can benefit greatly from a couple of things. 1) Bland-Altman plots with limits of agreement, so they can visualize the absolute error/bias and its spread, and 2) a regression equation for Fig 1, so that practitioners may "convert" Vertec to MyJump estimates and vice versa.
- There are some issues with reference formatting/names. Please ensure all citations are correct.

Introduction
- The introduction, although informative, is very lengthy and not all of the information is needed to provide the reader with context and a rationale. As such, I think many parts of the introduction can be pared down and streamlined. For example, the importance and use of the VJ, including in sport, can be reduced to one paragraph. To streamline things, I suggest "grouping" references to support more general statements.
L157–158 - Please refine this statement; for example, "reliability, in terms of absolute agreement between the measurement tools, would be low, with a detectable, systematic difference."

Methods
L167–169 - This sentence is difficult to read. I suggest creating a table with participant characteristics.
- Throughout, please change "weight" to "body mass" and "gender" to "sex"
L193 – Please rewrite this equation as (gt^2)/8, where g = 9.81 m/s^2. This is not only cleaner, but will allow the readers to better understand its origins rather than having a number to the hundred-thousandths place
- Please clarify that statistical assumptions are met; that is, normality and homoscedasticity

Discussion
L291 - Please remove "to the real world"
L331 - You can also add that you compared data from a single jump, so between-jump reliability or construct validity of their maximum jump height may not matter.

Experimental design

The focus on "lower limb power" in this study goes against a number of number of expert recommendations, in that vertical jump performance is not a measure of power, but rather is a measure of impulse (see below for references). I recommend dropping power from the paper.
https://www.ncbi.nlm.nih.gov/pubmed/19675467
https://www.ncbi.nlm.nih.gov/pubmed/26529527

Validity of the findings

no comment

Additional comments

no comment

·

Basic reporting

The text is written clearly and concisely – well done to the authors. The introduction shows good depth of knowledge of background research and published literature.

You make solid points in your discussion – I fully agree on your recommendations of using either My Jump or the Vertec. However, please provide regression equations from your data to estimate one from the other. This would provide readers & practitioners with an accurate equation to compare their population to others, even if they use different measurement tools (Vertec or My Jump).

A Bland Altman Plot would also provide a useful visual for readers to interpret the difference between Vertec & My Jump. Consider plotting the Vertec, and/or the average of Vertec & My Jump on the x-axis (1 or 2 plots).

Experimental design

In your introduction, you described differences in male and female athletes for vertical jump performance – you should in turn split your data into male and female samples and then run the same reliability analysis on the split groups.
I would also like to see if there are differences in reliability for higher and lower level athletes. All this data appears to be in your data bank, so should not require a huge amount of further analysis.

Validity of the findings

It appears you have 2 very large outliers in your data set. Are you sure these are correct – given how well related all other data points are, these seem very unique?

There are some missing data points in your data set – can you explain these? Do you think it is best to remove these participants altogether? For example, participant 88 has 2 jumps missing from the app. The difference between the participants max jumps is 15cm, yet the highest jump was achieved in a trial where there is no app data.

Additional comments

Lines 65-66 could read better. Recommend something like “key component for athleticism to determine performance…”
Line 76 – remove ‘the research’ prior to literature.
Line 86 – replace ‘last’ with ‘lastly’
Line 207 – replace ‘20s’ with ’20 seconds’.
Lines 209-210 – remove this sentence beginning with “All jump trials...”

---

## Round 0.2 · Minor Revisions

Dear authors.

One of the reviewers is recommending "minor revisions". These should be very easy to address. We hope to be able to accept your article once these revisions are completed.

·

Basic reporting

- I thank the authors for their thorough response and edits. I have (mostly) brief comments that I feel need to be addressed before the paper is ready for publication.
- The authors' response regarding the inclusion of "peak power" in this study is appreciated. However, beyond my concern regarding its validity for actually measuring power, there are pure statistical concerns with running ICCs on these measures. Specifically, a regression is used that is a function of both VJ height (differs between measures) and body mass (identical between measures). The constant offset (y-int) is not important as it has no associated variance. However, the body mass will inflate the variance identically in both measures, such that it will inflate the ICCs. That is, it will create shared variance. This can be shown both mathematically (using propagation of error and the ICC formulae) and pragmatically via simulation (example below).

#######################
# Simulation (R code for one example):
library(DescTools)
x1 <- rnorm(150,0,20)
x2 <- x1 + rnorm(150,0,20)
ICC(data.frame(x1,x2)) # shows ICC without a random offset added (akin to VJ)

# If we generate a random offset (akin to adding body mass)
x3 <- rnorm(150,0,30)
ICC(data.frame(x1+x3,x2+x3))
# Note the ICCs are much greater. How much they increase will be a function of the relative variances of x3 to x1 and x2. The larger the x3 variance, the more it will inflate the ICCs given x1 and x2 variance components.
########################

If you insist on keeping peak power (I defer to the editor here), then I think this is worth mentioning in your discussion or limitations: The use of a regression to estimate peak power inherently inflates ICC estimates due to the addition of shared variance.

Introduction
- This is much more succinct. Thank you.

Methods
- L149–151: I think you can get rid of this sentence and just reference Table 1 at the end of the sentence on L149. This would make things easier to read.
- L154: Please include the IRB study ID
- L220: Please add "systematic" before "differences"
- L221: Please add limits of agreement to the BA plots. Depending what they are, you may wish to discuss them when interpreting your results

Results
- L228: No need to redefine ICC and CI here
- L249: When referencing "statistical significance", could you please prefix "significant" with "statistically" (changing adverbial forms as appropriate)? This is to differentiate practical from statistical significance.

Discussion
- L308–309: What are "peak power" calculations an ideal measure of? Please clarify, bearing in mind the limitations of the measure.

Conclusion
- L339–343: You may wish to mention that the provided regressions can improve/correct for this bias

Figures
- Please add limits of agreement to the BA plots

Experimental design

n/a

Validity of the findings

n/a

Additional comments

n/a

·

Basic reporting

Thank you for the addition of regression equations should aid practitioners in comparing data, if needed.

Experimental design

The Bland-Altman plots really help visualise your findings, good work.

Validity of the findings

I have no strong preference - it is clear you have a logical process for cleaning data and the outliers must be from your population as you have pointed out. No need to further amend.

Additional comments

Thank you for addressing my previous comments.

---

## Round 0.3 · accepted · Accept

Dear Vanessa,

Thank you for your submission to PeerJ.

I am writing to inform you that your manuscript - The reliability of vertical jump tests between the Vertec and My Jump phone application - has been Accepted for publication. Congratulations!